# Fecal microbiota profiles of growing pigs and their relation to growth performance

Emilia König[1,2]*, Shea Beasley[3¤], Paulina Heponiemi[4], Sanni Kivinen[4], Jaakko Räkköläinen[4], Seppo Salminen[4], Maria Carmen Collado[4,5], Tuomas Borman[6], Leo Lahti[6], Virpi Piirainen[1,2], Anna Valros[2], Mari Heinonen[1,2]

**1** Faculty of Veterinary Medicine, Department of Production Animal Medicine, University of Helsinki, Helsinki, Finland, **2** Faculty of Veterinary Medicine, Department of Production Animal Medicine, Research Centre for Animal Welfare, University of Helsinki, Helsinki, Finland, **3** Vetcare Ltd., Mäntsälä, Finland, **4** Functional Foods Forum, University of Turku, Turku, Finland, **5** Department of Biotechnology, Institute of Agrochemistry and Food Technology–National Research Council (IATA-CSIC), Valencia, Spain, **6** Department of Computing, University of Turku, Turku, Finland

¤ Current address: Sheaps Oy, Ojakkala, Finland
* emilia.konig@helsinki.fi

**Data Availability Statement:** The 16S rRNA sequences for the current study along with a metadata file are available in the Harvard Dataverse Network (https://doi.org/10.7910/DVN/GHDU2U).

## Abstract

The early gut microbiota composition is fundamentally important for piglet health, affecting long-term microbiome development and immunity. In this study, the gut microbiota of post-parturient dams was compared with that of their offspring in three Finnish pig farms at three growth phases. The differences in fecal microbiota of three study development groups (Good, Poorly, and PrematureDeath) were analyzed at birth (initial exposure phase), weaning (transitional phase), and before slaughter (stable phase). Dam *Lactobacillaceae* abundance was lower than in piglets at birth. *Limosilactobacillus reuteri* and *Lactobacillus amylovorus* were dominantly expressed in dams and their offspring. Altogether 17 piglets (68%) were identified with *Lactobacillaceae* at the initial exposure phase, divided unevenly among the development groups: 85% of Good, 37.5% of Poorly, and 75% of Premature-Death pigs. The development group Good was identified with the highest microbial diversity, whereas the development group PrematureDeath had the lowest diversity. After weaning, the abundance and versatility of *Lactobacillaceae* in piglets diminished, shifting towards the microbiome of the dam. In conclusion, the fecal microbiota of pigs tends to develop towards a similar alpha and beta diversity despite development group and rearing environment.

## Introduction

In pigs, there are indications of early life microbial colonization within the immediate prenatal period [1]. The following postnatal days show an increase in microbial diversity and richness of the piglet gut microbiota [2–4].Undisturbed early life microbial exposure favors the development of a stable mature microbiota [5] and has a long-term influence on the later life of the host [4,6,7].

**Funding:** The Ministry of Agriculture and Forestry of Finland (Grant number: 504/03.01.02/2018; https://mmm.fi/etusivu), Vetcare Ltd. (Grant number: 031019; https://www.vetcare.fi/), and A-Farmers Ltd. (Grant number: 281019; https://www.atriatuottajat.fi/) funded the research project of M. H., including this study. Additionally, at the time this study was conducted, S.B. was an employee of Vetcare Ltd. (Grant number: 031019; https://www.vetcare.fi/). The specific roles of this author are articulated in the 'author contributions' section. Further, E.K. was funded with material grants for this study by the Finnish Veterinary Foundation (https://etts.fi/en/frontpage/) and the Finnish Foundation of Veterinary Research(https://www.sels.fi/). The Spanish National Plan for Scientific and Technical Research and Innovation funded the work of M.C.C. (CEX2021-001189-S/ MCIN/AEI /10.13039/501100011033, https://www.aei.gob.es/ ). Open access was funded by Helsinki University Library. The funders had no role in study design, data collection and analysis, decision to publish, or preparation of the manuscript.

**Competing interests:** Vetcare Ltd. (Grant number: 031019; https://www.vetcare.fi/), and A-Farmers Ltd. (Grant number: 281019; https://www.atriatuottajat.fi/) funded the research project of M. H., including this study. Further, at the time this study was conducted, S.B. was an employee of Vetcare Ltd. (Grant number: 031019; https://www.vetcare.fi/). This does not alter our adherence to PLOS ONE policies on sharing data and materials. There are no patents, products in development or marketed products associated with this research to declare.

Microbial development phases in young mammals, also used as sampling time points in this study, are presented in Fig 1 [8–10]. In pigs, the *initial exposure phase* incorporates the time span from fertilization to the end of lactation. During the *transitional phase*, the maturation of the microbiome begins after weaning, and the *stable phase* takes place during the finishing period [11,12].

To ensure microbial transfer to offspring during pregnancy, delivery, and lactation [1,8,11,13], the maternal microbiome shifts [14,15]. In sows, the gut microbiota changes towards a more diverse state through the progressing pregnancy and all the way until weaning [14]. Developing piglets' fecal microbiota is influenced by both their dams' microbiota and the pig-milk microbiota during the initial exposure phase [16,17] (Fig 1). Specific strains of *Bifidobacterium* and *Lactobacillaceae* are among the first microbes to colonize young mammals [18–20] and are indicated to be present in the immediate prenatal period [1].

Microbial research in pigs has focused mainly on improving feed efficiency and animal growth or improving herd health [5]. Whether there are differences in fecal microbiota related to the growth of pigs throughout their lives is not known. Our aim was to investigate the development of the pig gut microbiota from birth to slaughter in the following three development groups: Good, Poorly, and PrematureDeath pigs. Pigs were sampled at birth and again around weaning, and in the development groups Good and Poorly also prior to slaughter. We hypothesized that the fecal microbiota would differ between the development groups and sampling times and that the development group Good would have a richer and more diverse microbiota than the other two development groups. Secondly, we aimed to compare the gut microbiota of dams after parturition with that of their offspring throughout the piglets' lives. We hypothesized that differences would exist between the three development groups already in the fecal microbiota of their dams soon after parturition.

## Materials and methods

### Ethics statement

All farmers signed a written consent for their animals to be included in this study. The experimental protocol was approved by the Institutional Review Board of the Regional State Administrative Agency for Southern Finland (ESAVI/16950/2018).

### Farms, housing, and management

Three commercial piglet producing farms and their corresponding finishing farms (Farms 1–3) from south and southwestern Finland took part in this study. Piglet producing farms 1–3 had 1000, 1100, and 1271 crossbred Yorkshire\*Landrace sows, respectively. (S1 Table in supporting information shows detailed sow production numbers). Sows were inseminated with Duroc semen to produce three-breed piglet crosses. Each piglet producing farm delivered their piglets to two corresponding finishing farms with 2200 and 2300, 1100 and 3500, and 1500 and 1700 crossbred Yorkshire\*Landrace\*Duroc pigs for Farms 1–3 respectively.

Piglets were housed during lactation with their dams in farrowing pens (about 4.5 m$^2$), with partly slatted floors, and the dams in farrowing crates. No bedding was offered to piglets in the farrowing units, except for Farm 3, which provided piglets with a covered creep area with sawdust bedding. Dams received a small amount of straw before parturition. Between batches, pens were pressure washed, but when occupied by sows and piglets, the piglets had contact with dam fecal matter. Cross-fostering of piglets was managed by the caretakers and commonly practiced on the study farms to achieve suitable litter compositions and sizes for the sows and to assure that all piglets received milk.

| Initial exposure phase Days 114 pn to 28 pp | Transitional phase Days 29 to 80 | Stable phase Days 81 to 190 |
|---|---|---|
| - Maternal microbiome<br>- Lactation<br>- Medication exposure<br>- Environmental factors<br>- Hygiene & Disease | - Diet<br>- Host genes<br>- Medication exposure<br>- Environmental factors<br>- Hygiene & Disease | - Diet<br>- Medication exposure<br>- Environmental factors<br>- Hygiene &Disease |

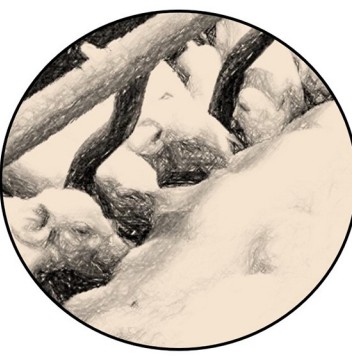 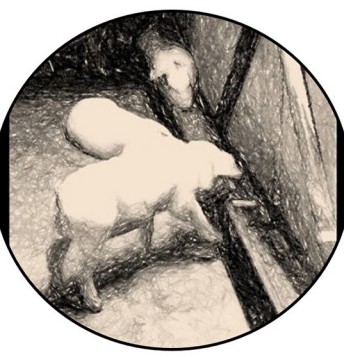 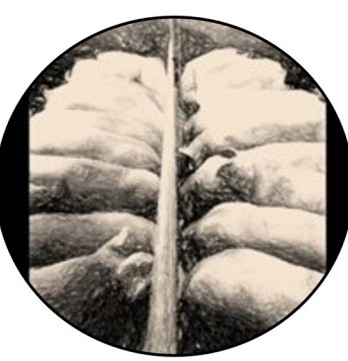

Firmicutes including Lactobacillaceae → Bacteroidetes

Bacterial diversity

**Fig 1. Schematic illustration of the development of the microbiome in pigs through the initial exposure, transitional, and stable phases, with phase related factors and indication of direction of change in bacterial phyla and diversity (modified from [8–10]).** pn = prenatal, pp = postpartum.

After weaning at approximately four weeks of age, piglets were moved into the weaning unit on the same farm. Groups of approximately 20 piglets per pen were housed together in the weaning unit. At around ten weeks of age, the pigs were moved to the finishing farms and kept in pens in groups of 20–25 pigs. In the weaning and finishing units, pens had partly slatted floors and a long feeding trough, and the pigs had a space allowance of about 0.4 m² for weaned pigs and 0.9 m² for finishing pigs. Chains and/or pieces of wood were provided for enrichment in both weaning and finishing units on all farms. When pigs were moved to new pens, sawdust was introduced. Thereafter, small amounts of sawdust, straw, or peat were added by caretakers.

All pigs were fed according to Finnish standards without any added microbes (Immonen, personal communication). On all farms homegrown grains (barley, wheat, oats) were mixed with a commercial protein source (barley-based protein feed or soy) and concentrate feed. All except suckling piglets were liquid fed and a mix of organic acids with a formic acid base was added to the liquid feed mixture for preservation. In addition to sow milk, suckling piglets received commercial feed, which was heat-treated (81°C) to eliminate possible contamination of feed and ensure the safety of piglets. Male pigs were surgically castrated during their first week of life and received pain medication intramuscularly after castration. Pigs were not tail

docked, nor were their teeth clipped or ground. No routine antimicrobials were used. All management procedures on the farms were performed by caretakers according to farm practice. Study pigs were housed together with non-study pigs throughout their lives.

### Study animals and selection criteria

All pigs born on the farms during the tracking week prior to Check-up 1 and the initial exposure phase (Fig 2) were individually ear tagged at birth. While 2366 pigs were born to 160 dams (S1 Table) only the three largest and three smallest pigs from each litter (n = 960) were regarded for this study and followed from birth to slaughter.

During the follow-up time, the pigs were checked five times on the farms. Fig 2 shows what data was recorded, collected, and evaluated from each check-up. The first check-up was during the initial exposure phase, when the piglets were 2–6 days old. During the birth week, the piglets were weighed, piglet feces samples were taken and the information on medication of dams and piglets were collected. The second check-up was during the transitional phase, when the piglets were 23–42 days old. At this point, piglet medication information and feces samples were collected. During the third and fourth check-ups, when the pigs were 64–83 and 99–123 days old, respectively, piglet medication information was collected. The final check-up was

**Fig 2. Collection of information and samples from study pigs and their dams during the five check-up visits to the farms and slaughterhouse.** Check-ups 1, 2, and 5 line up with fecal sampling during the initial exposure, transitional, and stable phases, respectively. Medication information was collected at all check-ups, piglets were weighed at first check-up, and meat inspection results were collected from the slaughterhouse. Days of fecal sampling are indicated with pig silhouettes.

**Table 1. Criteria used to classify study pigs into three development groups.**

| Development group | Pig feces sample availability | Dam feces sample availability | Growth | Dam antimicrobial medication 0–3 days prepartum | Piglet antimicrobial medication between birth and slaughter / death | Meat inspection findings |
|---|---|---|---|---|---|---|
| Good n = 13 | 3 | yes | ≥ Farm ADG + ½×SD | no | no | no |
| Poorly n = 8 | 3 | yes | ≤ Farm ADG– ½×SD | no* | NR | yes |
| Premature Death n = 4 | 2 | yes | NA | NR | NR | NA |

ADG = average daily gain, SD = standard deviation

* = ketoprofen was accepted, NA = not applicable, NR = can be yes or no, non-relevant for selection of study pigs.

during the stable phase when the pigs were 133–149 days old. The last fecal sample was collected along with medication information. The pigs were slaughtered at 150–191 days of age. At slaughter, carcass weight and routine meat inspection (MI) information, such as recorded findings of joint inflammation, abscesses, pneumonia, tail biting, or liver diseases, were collected. A more detailed description of timing and purpose of each check-up is presented in Fig 2.

All collected information on the pigs was used to create criteria for retrospectively forming three development groups and selecting samples for further analysis (Table 1). In total, 25 pigs from 24 litters met the criteria; 13 pigs were assigned to development group Good, 8 to development group Poorly, and 4 to development group PrematureDeath. The pigs from development group PrematureDeath died at the ages of 2, 28, 59, and 69 days. One pig died due to diarrhea; for the other three pigs, the reason for death is unknown.

All information of the selected 25 pigs is summarized in Table 2.

For organizational and biosecurity reasons, the pigs could not be sampled on exactly the same dates. Therefore, sampling dates are referred to as phases based on the microbiota development phase to which they correspond (Fig 1). The age of the pigs at each phase is shown in Fig 2. Fecal samples were collected from each pig three times: initial exposure phase, transitional phase, and stable phase (Fig 2). Transitional phase samples were collected during extensive dietary change from lactation to solid feed just after weaning. For three study pigs (12%; Table 2: Poorly8, PrematureDeath2, and PrematureDeath3), the samples in the transitional phase were collected before weaning. Stable phase samples were collected after microbial development was expected to have stabilized at the end of the finishing period. Dam samples were collected at the same time as piglet samples in the initial exposure phase.

## Data collection

The study pigs were weighed during their birth week and at slaughter to calculate an average daily gain (ADG) for the whole growing period (Table 2). Caretakers were asked to keep records of medications on separate forms prepared by the researchers. During each check-up, researchers collected the forms (Fig 2). Additionally, all pigs on all farms had extra ear-tags that were removed once a pig received an antimicrobial treatment course. The researchers checked the information from the collected medication forms with information on these ear-tags. Also dam medication data from three days prior to parturition were collected from farm records. Finally, MI information was obtained from the slaughterhouse.

**Table 2. Detailed information on study pigs of different development groups.** Average daily gain (ADG, grams) limits were calculated separately for each development group and each farm (for development group Good: Farm ADG+1/2xSD and for development group Poorly: Farm ADG-1/2xSD; development group PrematureDeath does not have an ADG because these pigs were neither weighed nor slaughtered).

| Farm | Pig development group and number | Number of fecal samples | ADG, g/day | Farm ADG limit, g/day | Pig antimicrobial medication | Meat inspection finding | Dam antimicrobial medication |
|---|---|---|---|---|---|---|---|
| 1 | Good1 | 3 | 778 | ≥ 761 | No | No | No |
| 2 | Good2 ŧ | 3 | 825 | ≥ 711 | No | No | No |
| 2 | Good3 | 3 | 780 | ≥ 711 | No | No | No |
| 3 | Good4[1*] ŧ f | 3 | 788 | ≥ 764 | No | No | No |
| 3 | Good5 ŧ | 3 | 808 | ≥ 764 | No | No | No |
| 3 | Good6 ŧ | 3 | 843 | ≥ 764 | No | No | No |
| 3 | Good7 | 3 | 780 | ≥ 764 | No | No | No |
| 3 | Good8 ŧ f | 3 | 784 | ≥764 | No | No | No |
| 3 | Good9 ŧ | 3 | 861 | ≥ 764 | No | No | No |
| 3 | Good10[1*] ŧ | 3 | 773 | ≥ 764 | No | No | No |
| 3 | Good11 | 3 | 837 | ≥ 764 | No | No | No |
| 3 | Good12 ŧ | 3 | 767 | ≥ 764 | No | No | No |
| 3 | Good13 ŧ | 3 | 855 | ≥ 764 | No | No | No |
| 1 | Poorly1 | 3 | 644 | ≤ 698 | No | Yes[2*,3*] | No[4*] |
| 2 | Poorly2 ŧ | 3 | 566 | ≤ 648 | No | No | No |
| 2 | Poorly3 | 3 | 536 | ≤ 648 | No | No | No |
| 3 | Poorly4 ŧ | 3 | 700 | ≤ 702 | Yes[5*] | Yes[2*] | No |
| 3 | Poorly5 ŧ f | 3 | 697 | ≤ 702 | Yes[5*] | Yes[2*] | No |
| 3 | Poorly6 ŧ | 3 | 668 | ≤ 702 | No | No | No |
| 3 | Poorly7 ŧ | 3 | 682 | ≤ 702 | No | No | No |
| 3 | Poorly8 ŧ | 3 | 613 | ≤ 702 | Yes[5*] | Yes[3*] | No |
| 1 | PrematureDeath1 | 2 | NA | NA | Yes[6*] | NA | No |
| 2 | PrematureDeath2 | 2 | NA | NA | No | NA | No |
| 3 | PrematureDeath3 | 2 | NA | NA | No | NA | Yes[7*] |
| 3 | PrematureDeath4 | 2 | NA | NA | Yes[7*] | NA | No |

ŧ = cross-fostered during birth week, f = female

[1*] = same dam, [2*] = pleuritis

[3*] = organ rejection

[4*] = ketoprofen had been given to the dam

[5*] = no information available on the antimicrobial used

[6*] = amoxicillin

[7*] = benzylpenicillin procaine.

All fecal samples were collected straight from the rectum with factory clean disposable vinyl gloves. Sterile cotton swabs were used to help collecting fecal samples from small piglets. Samples were placed immediately in sampling tubes, and the tubes were put in a Styrofoam box on ice bricks. The box was emptied every two hours into a -18°C freezer. The samples were transported in a freezer to the University of Helsinki laboratory within 24 hours and stored at -80°C until analysis.

### *Lactobacillaceae* isolation and microbiota profile by 16S rRNA amplicon sequencing

Fecal *Lactobacillaceae* abundance was determined as described elsewhere [21]. Briefly, ten-fold dilutions of fecal samples were plated on blood liver (BL) agar (Nissui Pharmaceutical, Tokyo,

Japan). Non-identical colonies were chosen to be incubated in Gifu Anaerobic Medium Broth (GAM broth; Nissui Pharmaceutical, Tokyo, Japan). The DNA of non-motile rod-shaped bacteria was isolated, and a colony PCR and a 16S PCR were used to identify *Lactobacillaceae*. Species were identified with 96% homology, comparing sequences with the National Center for Biotechnology Information's (NCBI) Basic Local Alignment Search Tool (BLAST) database.

Fecal microbiota profiling was performed by use of the 16S rRNA amplicon sequencing as follows: Fecal samples (0.100–0.125 g) were weighed into 2 ml screw cap tubes, and InviMag® Stool DNA Kit (Invitek Molecular GmbH, Berlin, Germany) was used according to the manufacturer's instructions for performing microbiota analysis. For the extraction, we used a FastPrep®-24 Sample Preparation System (M.P. Biomedical, Irvine, CA, USA), a Heraeus Pico 17 Centrifuge (Thermo Fisher Scientific Ltd., Osterode am Harz, Germany), and a King-Fisher™ Purification System Type 700 (Thermo Fisher Scientific Ltd., Vantaa, Finland). A NanoDrop 2000 UV-Vis Spectrophotometer (Thermo Fisher Scientific Ltd., Wilmington, DE, USA) was used to measure the DNA concentration of eluates.

To normalize DNA concentration for the library run, the concentration was measured with a Qubit® 2.0 Fluorometer (Life Technology, Carlsbad, CA, USA). The V3-V4 variable regions of the 16S rRNA gene were amplified with primers selected from Klindworth and colleagues [22] to perform the libraries as described elsewhere [23]. With the NextEra XT Index Kit (FC-131-2001) (Illumina, San Diego, CA, USA), a multiplexing step was conducted and a Bioanalyzer DNA 1000 chip (Agilent Technologies, Santa Clara, CA, USA) was used to measure the DNA quality of the library PCR product and to verify the size (expectation of Bioanalyzer trace is ~550 bp). A 2 x 300 bp paired-end run on MiSeq Illumina platform was performed according to the manufacturer's instructions to sequence the libraries. The sequences were pre-processed by customizing an established workflow [24] with the DADA2 [25] algorithm (R package dada2 [26]). Quality plots were manually investigated, and the sequences were truncated (forward 260; reverse 210) and trimmed from the left (base 19). Reads were discarded if they had more than two expected errors. Default settings were used otherwise. Chimeric sequences were removed. From the processed sequences, the Amplicon Sequence Variants (ASVs) were inferred and taxonomically assigned against the Silva database (v. 138.1) [27]. Data were converted into *TreeSummarizedExperiment* [28] container in R/Bioconductor for further analysis.

## Data handling and statistical analysis

Descriptive statistics were run for cultured *Lactobacillaceae*. The *mia* R package was used to conduct statistical analyses of community composition and diversity. Alpha diversity analysis was performed with Shannon index, and development group-wise differences in alpha diversity were estimated with Wilcoxon test; the paired Wilcoxon test was used for comparison between phases. Beta diversity, or community composition, was estimated with Bray-Curtis dissimilarity index and visualized with Principal Coordinates Analysis (R package *scater*); the significance of pairwise differences between development groups was quantified with permutational analysis of variance (PERMANOVA; *vegan::adonis*). Differential abundance analysis was performed at the genus level using ANCOM-BC [29,30]. The *miaTime* package was used for time series analysis. The *ggplot2* and *miaViz* packages were the main packages for data visualization in R. The analyses were done with R-4.1.0 and Bioconductor 3.13.

## Results

Altogether 71 fecal samples from 25 growing pigs as well as 24 fecal samples from their 24 birth dams were analyzed. Due to cross-fostering, piglets were not always nursed by their birth

dams. In general, medication information was collected throughout the follow-up, but no pigs were medicated after Check-up 3 (Fig 2).

## Culturing of *Lactobacillaceae* members

The abundance of identified *Lactobacillaceae* strains at different phases is shown in Fig 3 for all three development groups at initial exposure and transitional phases, and also for development groups Good and Poorly at the stable phase. For the dams, *Lactobacillaceae* abundance is shown only at the initial exposure phase of the piglets. The highest abundance was found in development group Good at initial exposure phase and the lowest abundance in dams at the same phase.

A total of five *Lactobacillaceae* were isolated in seven dams out of 24. These were in descending order of prevalence: *Limosilactobacillus reuteri*, *Lactobacillus amylovorus*, *Lentilactobacillus parabuchneri*, *Lactobacillus rhamnosus*, and *Limosilactobacillus vaginalis*. Thirteen different *Lactobacillaceae* species were identified in the samples of growing pigs listed here in descending order of prevalence: *L. reuteri*, *L. amylovorus*, *L. vaginalis*, *Lactobacillus johnsonii*, *Lactobacillus ruminis*, *Ligilactobacillus salivarius*, *L. delbrueckii*, *Lactobacillus acidophilus*, *Limosilactobacillus mucosae*, *Lactobacillus murinus*, *Lactobacillus oris*, *Limosilactobacillus*

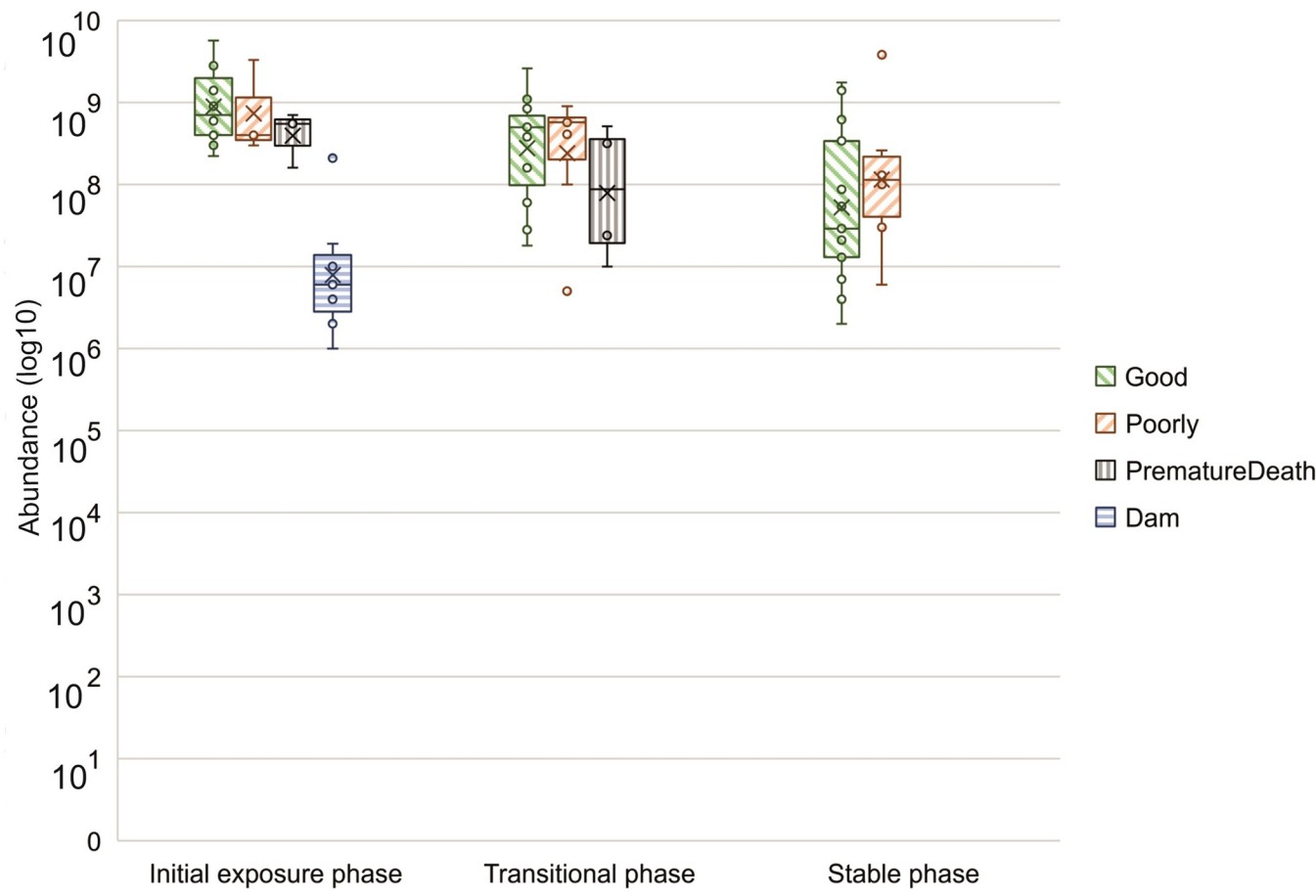

**Fig 3. Abundance (log10) of identified cultured *Lactobacillaceae* for development groups Good, Poorly, and PrematureDeath as well as the dams at all relevant phases (Initial exposure phase = 2–6 days, Transitional phase = 23–42 days, Stable phase = 133–149 days).** Minimum and maximum values are shown by whiskers, dots show outliers, the middle line shows the median and x shows the mean. Dams were not sampled at the transitional or stable phases.

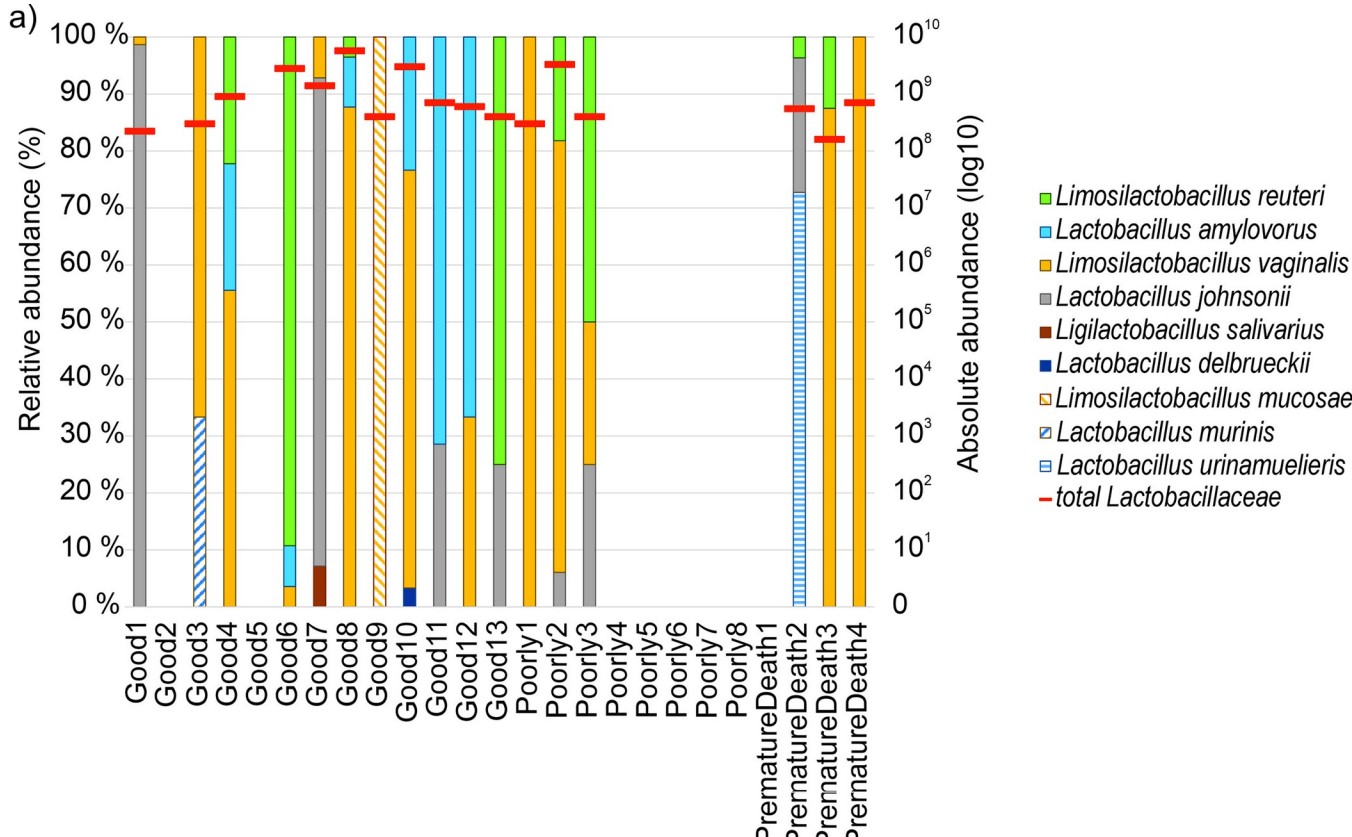

**Fig 4. Microbial abundances of *Lactobacillaceae* in pig fecal samples.** Samples were taken in the initial exposure phase (age 2–6 days) from development groups Good (n = 13), Poorly (n = 8), and PrematureDeath (n = 4). On the left y-axis relative abundance (%) and on the right y-axis total abundance (log10) of all cultured and identified *Lactobacillaceae* indicated with red horizontal bars for each sample over the relative abundance bars, respectively.

*pontis*, and *Lactobacillus urinamulieris*. Figs 4–6 summarize the species found in different piglet samples at the specific phases and the total abundance (log10) of cultured and identified *Lactobacillaceae*.

In the development group Good, in pig Good8 no culturable *Lactobacillaceae* were detected in the transitional phase sample. The pig was cross-fostered during the birth week but not medicated, and the transitional phase sample was taken after weaning. *Lactobacillaceae* were not detectable in pig Good2 at initial exposure and transitional phase but *L. reuteri* was present at the stable phase. Good2 had not been medicated but had been cross-fostered. In the development group Poorly, three of five pigs without *Lactobacillaceae* findings had MI findings (Table 2). The pig Poorly1 identified with *L. vaginalis* had a medicated dam as well as MI findings (Fig 4). Neither pig PrematureDeath2 nor its dam was exposed to medication (Table 2), and the pig expressed some diversity of three *Lactobacillaceae* species (Fig 4). The other three PrematureDeath pigs and/or their dams were exposed to medication (Table 2), which was reflected in low or no diversity of *Lactobacillaceae* in these pigs at the initial exposure phase (Fig 4). The average numbers of expressed *Lactobacillaceae* species for each development group and for cross-fostered and not cross-fostered animals are shown in Table 3.

The relative abundance of *L. reuteri* and *L. amylovorus* in individual growing pigs increased with time from birth to slaughter. Especially in Good pigs, the abundances of *L. reuteri* and *L. amylovorus* became stable earlier than in the other development groups (Fig 4–6).

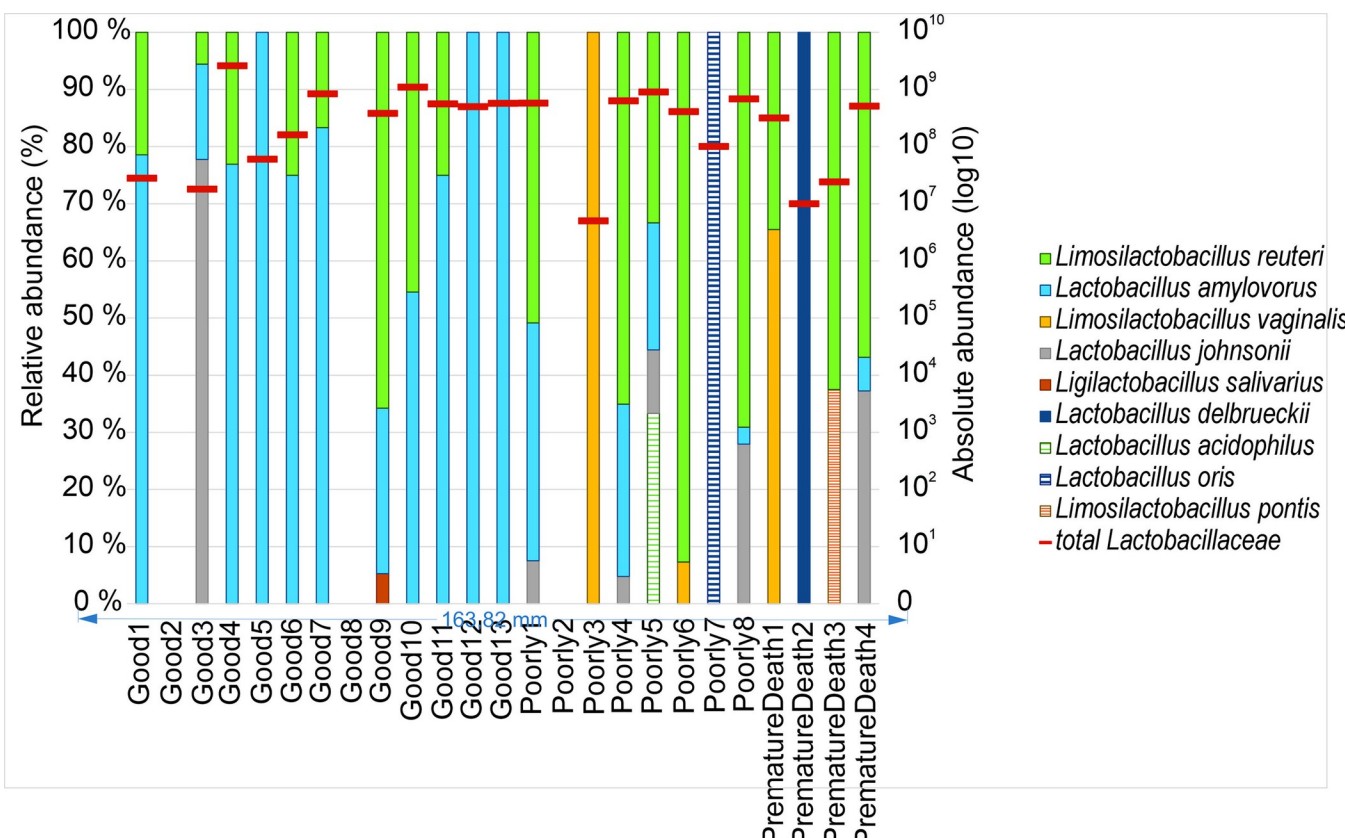

**Fig 5. Microbial abundances of *Lactobacillaceae* in pig fecal samples.** Samples were taken the transitional phase (age 23–42 days) from development groups Good (n = 13), Poorly (n = 8), and PrematureDeath (n = 4). On the left y-axis relative abundance (%) and on the right y-axis total abundance (log10) of all cultured and identified *Lactobacillaceae* indicated with red horizontal bars for each sample over the relative abundance bars, respectively.

Regarding *L. vaginalis*, the abundance of bacteria in the samples and the number of pigs expressing *L. vaginalis* decreased with time beneath detection. Similarly, the abundance of *L. johnsonii* and the number of pigs expressing the bacteria decreased with time, not, however, to non-detectable. By contrast, the number of pigs expressing *L. ruminis* increased with time from none to three. *L. salivarius*, *L. delbrueckii*, *L. acidophilus*, *L. mucosae*, *L. murinis*, *L. oris*, *L. pontis*, and *L. urinamulieris* were only expressed transiently in less than three pigs at varying phases.

Three *Lactobacillaceae* species were found in both sows and piglets, but only *L. reuteri* and *L. amylovorus* were expressed by both dams and their offspring. On Farm 3, five piglets expressed the same *Lactobacillaceae* species as their dams at different phases (Fig 7). Two piglets expressed the same *Lactobacillaceae* at the initial exposure phase, with decreasing abundance towards the stable phase. The other three expressed the same *Lactobacillaceae* at the transitional phase and the stable phase. All five of the piglets had been cross-fostered during the suckling period. Two of the study pigs with the same *Lactobacillaceae* species as in their dams' fecal samples had not been medicated (pigs Good6 and Good13; Table 2), and three had been both medicated and had MI findings (pigs Poor4, Poor5, and Poor8; Table 2). All of the other study pigs except for Poorly2 and PrematureDeath2 expressed either *L. amylovorus* or *L. reuteri* or both after weaning (Figs 5 and 6), but their dams did not.

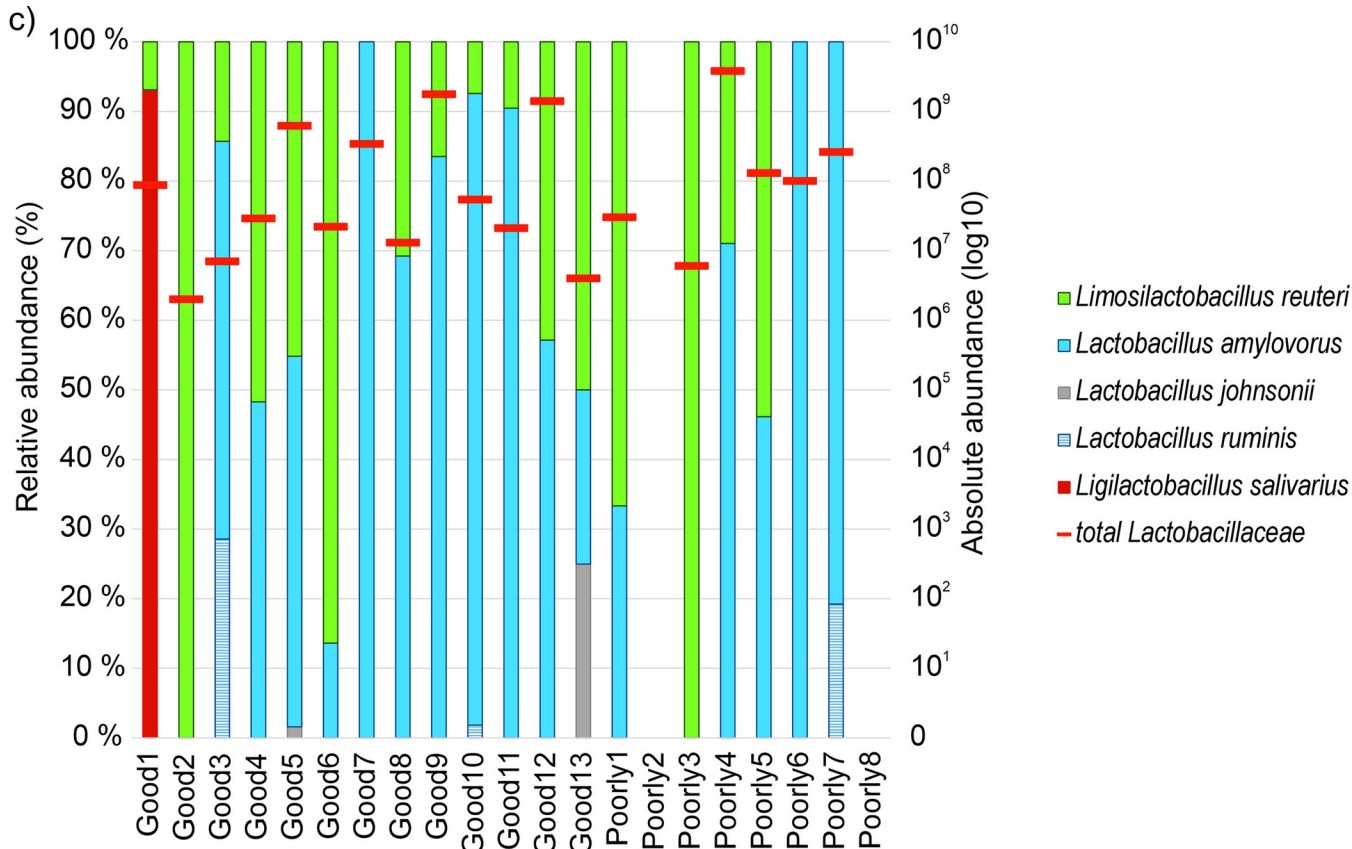

**Fig 6. Microbial abundances of *Lactobacillaceae* in pig fecal samples.** Samples were taken in the stable phase (age 133–149 days) from development groups Good (n = 13) and Poorly (n = 8). On the left y-axis relative abundance (%) and on the right y-axis total abundance (log10) of all cultured and identified *Lactobacillaceae* indicated with red horizontal bars for each sample over the relative abundance bars, respectively.

## Microbiota profiling by 16S rRNA amplicon sequencing analysis

Regardless of microbial seeding and environmental factors, the microbiota of pigs seemed to develop similarly. The beta diversity analysis in Fig 8 showed heterogeneity among samples at the initial exposure phase. With the passage of time, the microbiota converged, with the samples becoming more homogenic towards the end of the follow-up (Fig 8). The driver genera of the principal component axes 1 and 2 are shown in the supporting information (S1 Fig).

**Table 3. Average number of *Lactobacillaceae* species expressed in dams, in development groups good (n = 13), poorly (n = 8), and prematuredeath (n = 4), as well as in cross-fostered (n = 15) and not cross-fostered (n = 10) pigs at each phase.**

| Development group | Initial exposure phase | Transitional phase | Stable phase |
|---|---|---|---|
| Dams | 0.38 (range 0–3) | NS | NS |
| Good | 2.00 (range 0–3) | 1.62 (range 0–3) | 2.15 (range 1–3) |
| Poorly | 0.88 (range 0–3) | 2.13 (range 0–4) | 1.25 (range 0–2) |
| PrematureDead | 1.50 (range 0–3) | 2.00 (range 1–3) | NA |
| Cross-fostered | 1.3 (range 0–3) | 1.7 (range 0–4) | 1.8 (range 0–3) |
| Not cross-fostered | 1.9 (range 0–3) | 2.1 (range 1–3) | 2.0 (range 1–4)* |

NS = not sampled, NA = not available

* n = 6 because PrematureDeath pigs were already dead.

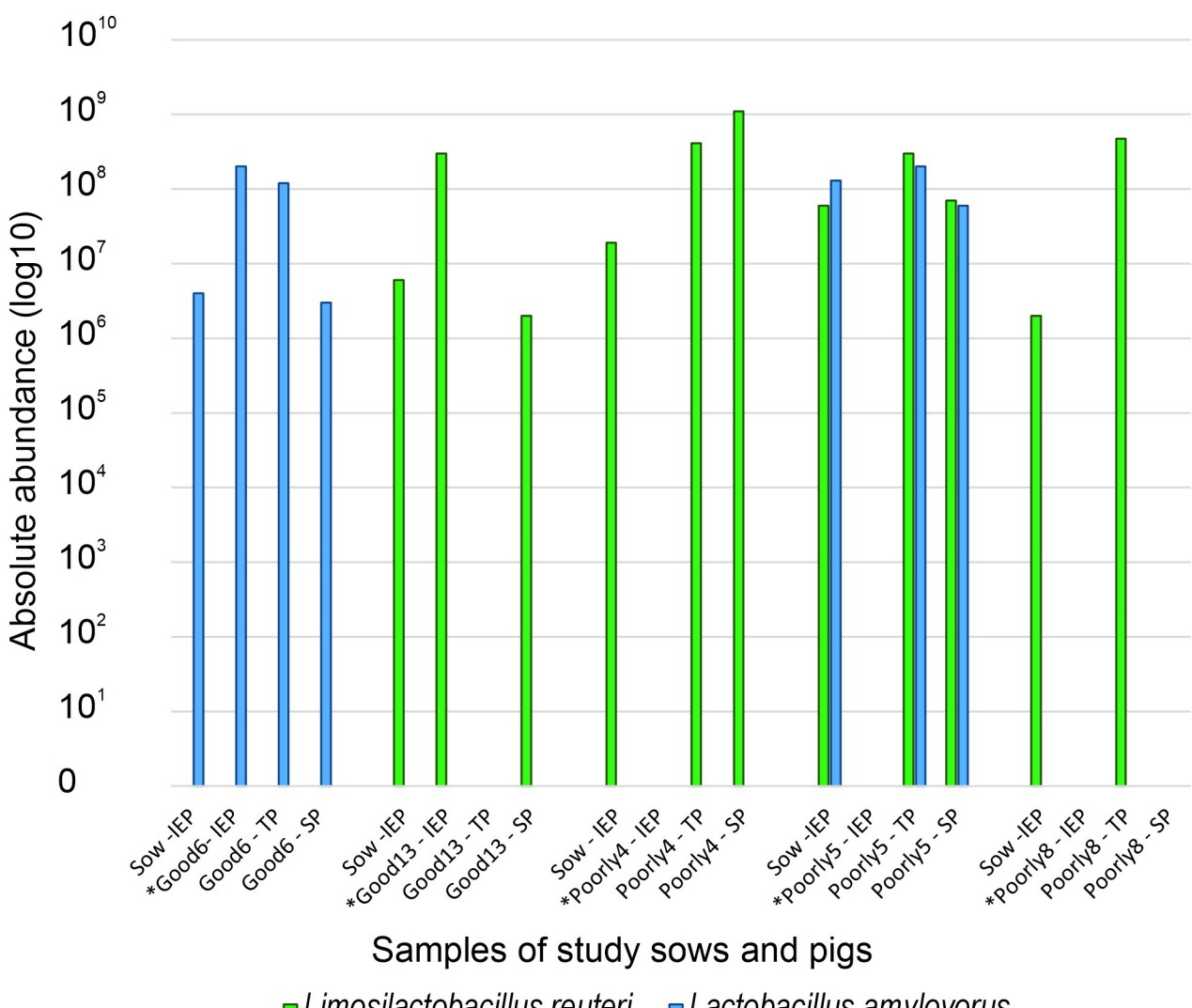

**Fig 7. Absolute microbial abundance of *Lactobacillaceae* (log10) in dams and their piglets.** Five study pigs expressed the same *Lactobacillaceae* as their dams at different phases (IEP = initial exposure phase, TP = transitional phase, SP = stable phase). * = pigs cross-fostered during birth week.

A piglet's microbiota proved to be very different from its dam's microbiota after birth but became more similar with time (Fig 9).

The beta diversity analysis (using Bray-Curtis distance) showed that the microbiota composition in all development groups differed numerically from the baseline right after birth, reaching the greatest difference in the sample around weaning, and staying at the same level until slaughter (S2 Fig).

Alpha diversity analysis revealed pigs from the development group PrematureDeath to have the numerically lowest average microbial alpha diversity relative to the other two development groups; however, differences between development groups were not significant. Looking at the initial exposure phase, the development group Good had the highest microbial diversity and the development group PrematureDeath the lowest (Fig 10). At the transitional phase, the development group PrematureDeath showed marked variation, whereas the microbial diversity of the development group Good was still higher than that of the development group

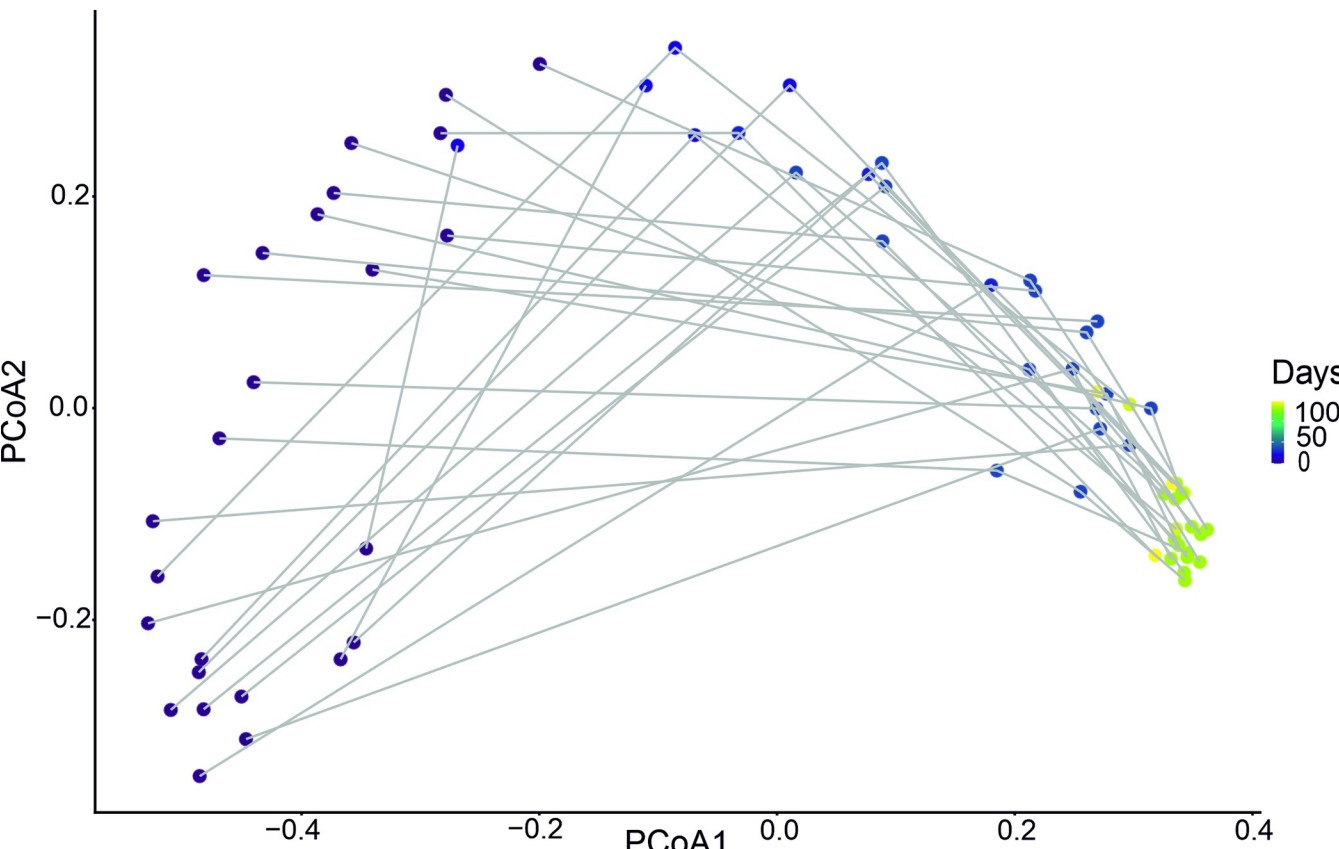

**Fig 8. Principal Coordinates Analysis (PCoA) with Bray-Curtis dissimilarity distance showing how samples from all study pigs across the development groups develop from a scattered group over time towards the same endpoint.** The convergence expressly relates to gut microbiome (dis)similarity. Change in color of dots expresses passing of time.

Poorly. At the stable phase, this was altered, as the microbial diversities decreased, but less in the development group Poorly than in the development group Good (Fig 10). The differences in alpha diversity over time were significant at almost all time points for all development groups (Fig 10). For the development group Good, a significant difference was seen between

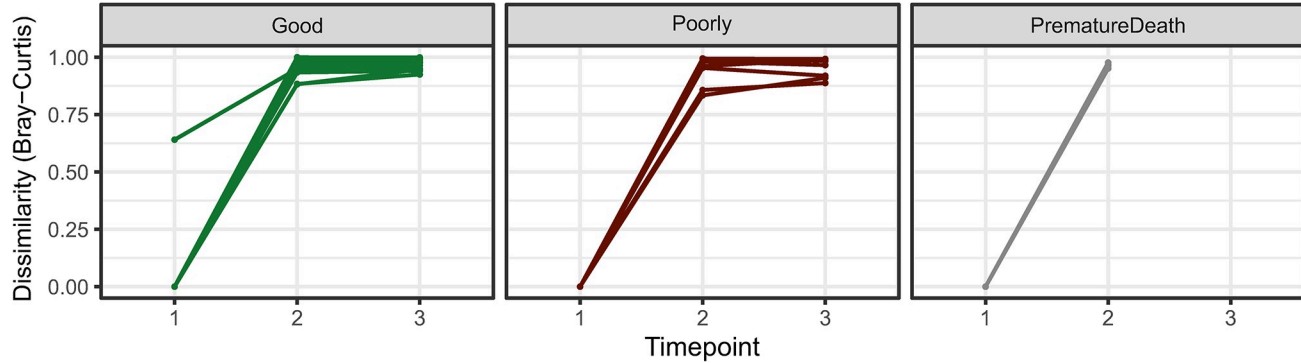

**Fig 9. Comparison of gut microbiome (dis)similarity of the development groups (Good n = 13, Poorly n = 8, and PrematureDeath n = 4) at three phases (1 = initial exposure phase, 2 = transitional phase, 3 = stable phase) to their dams' samples taken at the same time as the initial exposure phase samples of the piglets.**

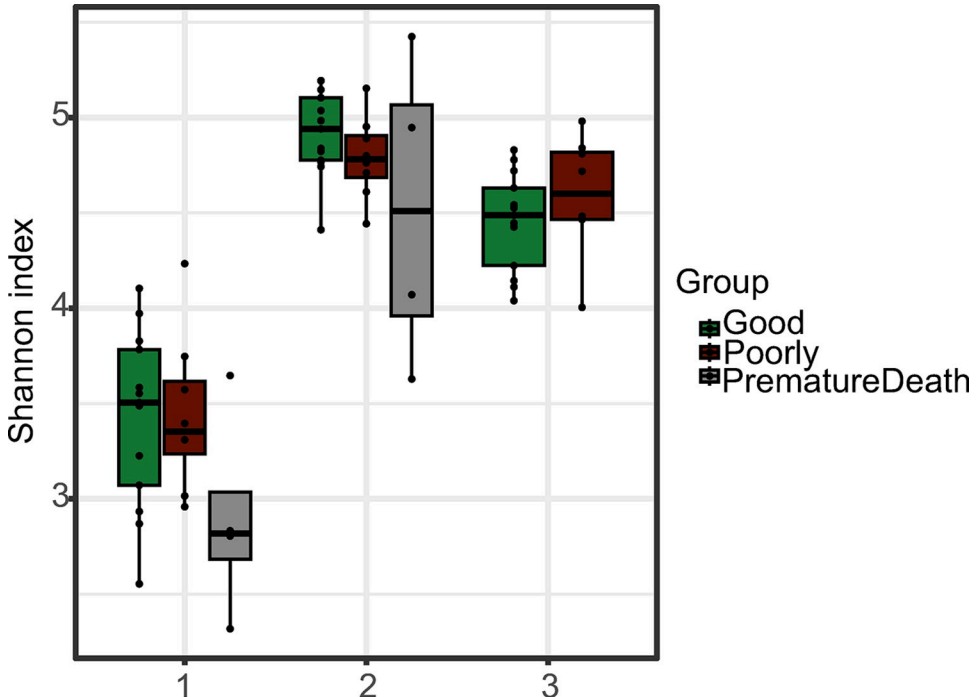

**Fig 10. Alpha diversity (Shannon index).** Samples divided into development groups (Good n = 13, Poorly n = 8, PrematureDeath n = 4) at three time periods (1 = initial exposure phase, 2–6 days; 2 = transitional phase, 23–42 days; 3 = stable phase, 133–149 days).

initial exposure phase and transitional phase ($P$ = 0.000), initial exposure phase and stable phase ($P$ = 0.002), and transitional phase and stable phase ($P$ = 0.02). For the development group Poorly, a significant difference was seen between initial exposure phase and transitional phase ($P$ = 0.001) as well as between initial exposure phase and stable phase ($P$ = 0.004). For the development group PrematureDeath, a significant difference was seen between initial exposure phase and transitional phase ($P$ = 0.04). Neither alpha nor beta diversity (S3 Fig) analyses showed differences between the dams of the pigs from the different development groups.

## Discussion

In this study, piglets at birth were identified with higher *Lactobacillaceae* abundance than their dams. As *Lactobacillaceae* are among the first early life microbes and have been shown to be the most abundant group of bacteria in pigs during the first three weeks of life [31]. *Lactobacillaceae* are mainly associated with maternal exposure [32], it is interesting that only 7 (29%) of the dams were identified with *Lactobacillaceae*. Although unexpected, this is in line with the results of Berry and colleagues [33] who reported numerous *Lactobacillaceae* species in piglets but none in sows. Of the seven sows identified with *Lactobacillaceae* in this study, five had mutual *Lactobacillaceae* with their offspring, mainly *L. amylovorus* and *L. reuteri*. These piglets were equipped with covered creep areas with sawdust bedding, keeping them warm in the farrowing pens and possibly facilitating microbiome development [34]. At lower environmental temperatures, overall alpha diversity has been shown to decrease and certain gut species become undetectable in several animal species, including swine [35]. In sows, as in other maternal individuals across animal species, fecal microbial alpha diversity typically increases

during pregnancy and until weaning, ensuring sufficient nutrient supply to the offspring during pregnancy and lactation [14,15,33]. Although the transfer of culturable *Lactobacillaceae* to piglets appeared fairly low in this study, the abundance of *Lactobacillaceae* in piglets was at its peak in the initial exposure phase. *Lactobacillaceae* are a core part of host health and have the ability to balance the intestinal microbiota via competitive exclusion [6,36,37]. Chen and colleagues [38] speculated these early settlers to pave the road for sow's fecal microbiota to colonize the newborn piglet's gut. Chen and colleagues [3] also reported microbiota in sow milk, the nipple surface, and slatted floors to tend to be similar to the piglet fecal microbiota. Most of the neonate piglets in our study experienced cross-fostering during the first days of life to have access to sufficient milk. Cross-fostering has been suggested to alter the neonate microbiome in cross-fostered siblings [5,39] and to elicit differences in their immune status and colonic microbiota [40]. Our study seems to support this as cross-fostered pigs expressed fewer species of *Lactobacillaceae* throughout their lives compared to non-cross-fostered individuals.

Altogether 17 piglets (68%) were identified with *Lactobacillaceae* at the initial exposure phase, divided unevenly among the development groups: 85% of Good, 37.5% of Poorly, and 75% of PrematureDeath pigs. The development group Good appeared to have more *Lactobacillaceae* diversity than the other two development groups. Of the dam-piglet paired *Lactobacillaceae* abundance, *L. reuteri* and *L. amylovorus* were the most prevalent in the development group Good. Contrary to our results, Morissette and colleagues [41] described high weight gain piglets to have lower proportions of *L. amylovorus* during the first two weeks of lactation than low weight gain piglets. This may be due to geographic differences in pig rearing, feed, and the environment may impact on the selection of the microbial diversity and therefore effect the abundancy of *L. amylovorus*. The high weight gain is thought to be linked to higher colostrum and milk intake, which is reported to affect the gut microbiome development [42]. In this study, during lactation, *L. reuteri* and *L. amylovorus* were detected in all development groups.

The number and diversity of *Lactobacillaceae* decreased in piglets between birth and weaning. *Lactobacillaceae* changes were seen in *L. mucosae*, *L. murinus*, and *L. urinamulieris* being outgrown by *L. acidophilus*, *L. oris*, and *L. pontis*. Cheng and colleagues [43] as well as Gaukroger and colleagues [44] reported the level of microbial diversity to be the lowest at the beginning of lactation on day 3, seen also in this study. *Lactobacillaceae* diversity and abundance were highest in the development group Good. In the development group Poorly, pigs with MI findings were medicated or their dams had been medicated prior to birth. Sow and pig medication also appeared to affect *Lactobacillaceae* diversity in PrematureDeath pigs. This indicates the negative effect of sow and piglet medication or their need to be medicated on development of the piglet microbiome. Piglets with low early life microbial abundance are likely to miss pivotal developmental windows [45]. Also, instability of the microbiota during early life renders the community structure of microbiota during this time more sensitive to environmental incursions [46] related to adulthood health and lifespan together with early life nutrition [47].

After weaning, the abundance and versatility of *Lactobacillaceae* in piglets decreased. This phenomenon is seen in total fecal microbiota as well, where piglet fecal microbiota starts to resemble dam fecal microbiota. The maternal innate effect influencing the piglet fecal microbiota has been reported to diminish after the first 93 days [48]. The same group reported microbial richness and diversity to decrease towards the end of the finishing period [48]. Age groups differed more from each other than from groups on different farms [49]. In this study, *L. amylovorus* and *L. reuteri* were most abundant at the age of 6 months regardless of the development group. Similar growth environments between farms appear to shape pig fecal microbiota composition within an age group. In addition, the microbiome is affected by co-

habitants [32,50]. Individuals receive and share microbes, both beneficial and pathogenic, with their frequent co-habitants [50,51], highlighting the importance of a healthy microbiome throughout the herd. Transplantation of gut microbiota can result in transfer of a disease phenotype [52], leading to speculation about whether a medication-originated dysbiosis affects co-habitants through transferring pathogens. Nowland and colleagues [53] found piglet microbial development and piglet growth to be improved by removing sow feces from the pen. In this study, the sows and piglets were in continuous contact with sows fecal matter while occupying pens. The rearing environment of pigs does not seem to support diversity, as the conditions are standardized for pigs to develop in a similar way. Feeding, medication, rearing environment, and use of enrichment are fairly similar on all farms. This makes large-scale production easier and the product more uniform but might pose challenges for the development of a diverse gut microbiota.

In our study, total microbial diversity and richness were higher and remained stable during the finishing period, consistent with reports by Knecht and colleagues [32] and Le Sciellour and colleagues [54]. Pigs in the development group Good expressed a healthy microbial colonization and did not require medication during their lives, nor did they have MI findings after slaughter. The development group Poorly showed stable but slow development of gut microbial diversity. The high microbial diversity close to slaughter age exceeds the level of the development group Good. Of the four pigs with MI findings in the development group Poorly, three pigs received medication and one pig had a medicated dam. These findings suggest a link between medication, gut microbiota, and reduced health.

A limiting factor for this study was the small sample size. In addition, intensive cross-fostering within a few days of birth complicated the comparison of microbial development. Cross-fostering transiently reshapes preweaning colon microbiota and immune status [40], and it brings a major confounding element to the study. Also, dietary details were not available. The commercial feed did not contain added microbes, but differences may have been present in feed composition between farms. However, early-life microbiome, including *Lactobacillaceae*, received from the dam shapes the core microbiome of pigs. When undisturbed, the diversity of the core microbiome remains throughout the animal's life. However, diet and the environment influence the microbiota in short term. In this study, pigs received the same diet throughout each farm and therefore the effect on the microbiome is constant.

## Conclusions

Piglets at birth were identified with thirteen different *Lactobacillaceae* species and with higher *Lactobacillaceae* diversity and abundance than in their dams. Of the seven sows identified with *Lactobacillaceae*, five had mutual *Lactobacillaceae* with their offspring, mainly *L. amylovorus* and *L. reuteri*. Pigs in the development group Good were not medicated and had no MI findings. These pigs expressed higher relative abundance of *Lactobacillaceae* during the initial exposure phase than pigs growing poorly or dying prematurely. In the development group Good, more diversity in total microbiota between birth and weaning was recorded, indicating that these pigs expressed healthy microbial colonization. Nevertheless, a similar fecal microbial development was seen in alpha and beta diversity of the pigs, regardless of the pig's development group or rearing farm.

## Supporting information

**S1 Fig. Driver genera of principal component axes 1 (PC1) and 2 (PC2) of the Principal Coordinates Analysis in Fig 8 in Results.** The most correlated genera are listed and their correlations with the axes are shown. A human keystone bacterium Christensenella *minuta* is

studied for reducing obesity in humans. *Akkermansia muciniphila* is used to treat obesity and diabetes in humans.
(DOCX)

**S2 Fig. Comparison of gut microbiome (dis)similarity of the development groups (Good n = 13, Poorly n = 8, and PrematureDeath n = 4).** Samples taken during the transitional and stable phases differ numerically from the initial exposure phase sample.
(DOCX)

**S3 Fig. Comparison of dam samples.** The a) alpha diversity and b) beta diversity analyses of dams of the pigs from the different development groups (Good, Poorly, and PrematureDeath) relating to Fig 10 in Results.
(DOCX)

**S1 Table. Production information of sows giving birth during tracking week for all study farms.** All parturitions on the farms were tracked for one week and all piglets born during the week were ear tagged for individual identification.
(DOCX)

## Acknowledgments

Sini Marttila is thanked for invaluable help on the farms.

## Author Contributions

**Conceptualization:** Emilia König, Shea Beasley, Seppo Salminen, Anna Valros, Mari Heinonen.

**Data curation:** Emilia König, Maria Carmen Collado, Tuomas Borman, Leo Lahti.

**Formal analysis:** Tuomas Borman, Leo Lahti.

**Funding acquisition:** Emilia König, Mari Heinonen.

**Investigation:** Emilia König, Shea Beasley, Virpi Piirainen, Anna Valros, Mari Heinonen.

**Methodology:** Emilia König, Tuomas Borman, Leo Lahti, Mari Heinonen.

**Project administration:** Emilia König, Mari Heinonen.

**Resources:** Emilia König, Shea Beasley, Paulina Heponiemi, Sanni Kivinen, Jaakko Räkköläinen, Maria Carmen Collado, Virpi Piirainen, Anna Valros, Mari Heinonen.

**Software:** Tuomas Borman, Leo Lahti.

**Supervision:** Mari Heinonen.

**Validation:** Shea Beasley, Seppo Salminen, Leo Lahti, Anna Valros, Mari Heinonen.

**Visualization:** Emilia König, Shea Beasley, Tuomas Borman, Leo Lahti.

**Writing – original draft:** Emilia König, Shea Beasley.

**Writing – review & editing:** Emilia König, Shea Beasley, Paulina Heponiemi, Sanni Kivinen, Jaakko Räkköläinen, Seppo Salminen, Maria Carmen Collado, Tuomas Borman, Leo Lahti, Virpi Piirainen, Anna Valros, Mari Heinonen.

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
