## [Decision Letter · Decision Letter 0]

14 Feb 2024

PONE-D-24-03235Fecal microbiota profiles of growing pigs and their relation to growth performancePLOS ONE

Dear Dr. König,

Thank you for submitting your manuscript to PLOS ONE. After careful consideration, we feel that it has merit but does not fully meet PLOS ONE’s publication criteria as it currently stands. Therefore, we invite you to submit a revised version of the manuscript that addresses the points raised during the review process.

Please provide much more details about the experiment to allow the reviewers to determine if the data is appropriately sound for publication in PloS one. Make sure to address all comments exhaustively. 

We look forward to receiving your revised manuscript.

Kind regards,

Franck Carbonero, PhD

Academic Editor

PLOS ONE

Journal Requirements:

"I have read the journal's policy and the authors of this manuscript have the following competing interests:

at the time this study was conducted, S.B. was an employee of Vetcare Ltd.

Other authors have declared that no competing interests exist."

Additional Editor Comments:

Please provide much more details about the experiment to allow the reviewers to determine if the data is appropriately sound for publication in PloS one. Make sure to address all comments exhaustively.

Reviewers' comments:

Reviewer's Responses to Questions

**Comments to the Author**

1. Is the manuscript technically sound, and do the data support the conclusions?

Reviewer #1: No

Reviewer #2: Partly

2. Has the statistical analysis been performed appropriately and rigorously? 

Reviewer #1: No

Reviewer #2: Yes

3. Have the authors made all data underlying the findings in their manuscript fully available?

Reviewer #1: Yes

Reviewer #2: No

4. Is the manuscript presented in an intelligible fashion and written in standard English?

Reviewer #1: Yes

Reviewer #2: No

5. Review Comments to the Author

Reviewer #1: The manuscript attempts to describe an evaluation of microbial diversity in growing pigs with different growth performance. However, there are many shortcomings with the manuscript that need to be addressed and considered before further evaluation of the manuscript.

Lines 50-102: The Introduction should be shortened, targeted, and concluded with focused objective statements.

Lines 61-65: should be moved to Discussion

Lines 76-84: should be moved to Discussion

Line 118: How many piglets per site? What were the sow production numbers (e.g. total born, still born)?

Line 127: How far were piglets transported?

Line 128: How many total pens in the weaning and finishing units?

Line 129: Did one weaning unit house all the pigs?

Line 129: Were the finishing farm groups similar to the weaning unit groups or completely random?

Line 137: Was feed tested for microbial contribution to the results?

Line 138: What was the method and purpose of heat treatment?

Line 147: Is it correct that only ~5 piglets were born to each sow? Further on, only 25 piglets are considered, so the discrepancy must be explained.

Line 148: How were pigs selected and identified?

Line 150: What time points were evaluated?

Line 151: Birth weight and current weight?

Line 152: Were pigs selected for and marketed based on days on feed or body weight?

Line 154: Provide some (not all) details in the text for the reader.

Line 161: insert "during lactation" after "information"

Line 162: delete "These criteria are summarized in"

Line 163: Why were only these pigs selected?

Line 163: Does that mean at least one pig from each litter?

Line 165: When did these pigs die?

Line 165: Did you confirm the reason was injury?

Line 175: How are the sampling dates considered in the statistical analysis?

Line 253: There is no indication how the potential influence of different farms was considered in the analysis. How was each farm tested as a separate block? How was parity influence considered? What are the performance metrics considered and how were they evaluated?

Line 306: Was this pig considered in further analysis?

Line 401: What are other probable sources?

Line 418: Were any surfaces swabbed in this study? How were the rooms cleaned and disinfected before farrowing?

Line 418: Why were cross-fostered pigs chosen and not excluded?

Line 444: Why were medicated pigs chosen and not excluded?

Lines 486-487: This is a major shortcoming of the manuscript. Diet composition is known to influence the microbiome. Lack of information such draws serious questions about the validity of the study results.

Figures: All figures should include all the information needed to stand alone.

Reviewer #2: Author studied Fecal microbiota profiles of growing pigs and their relation to growth performance.

Line 113: since there were only three farms, please note the number of sows and finisher pigs from each farm.

Line 118 please provide the size of pen.

Line 122 “The pens were never completely clean.”

Were pens cleaned and disinfected before introducing the next group of sows?

Were sows from all 3 farms received the straw before parturition?

Line 123 What was the rule used for crossfoster?

Line 132 to 135: Were these practices true for all 3 farms?

Line 138: Is there scientific evidence that can confirm no probiotics are being added in any stage of production from all three farms? Since diets have greatly impacted the microbiomes of pigs, please provide diet compositions.

Line 147: What were the pig's selection criteria? How many piglets per dam were selected? What is the dam distribution for each farm?

Line 163: How was the gender distribution between the three categories and the number of pigs distributed from each farm?

The number of observations for the Premature death group is very low. At which stage of production were these 4 pigs dead?

Table 1: Please change “½*SD” to “1/2×SD”.

Line 179: Were samples for the transitional phase occurring on the date of weaning? Please clarify.

Table 2: Please provide units for noncategorical traits.

Line 208: Swab samples were collected from pigs at younger ages, while fecal samples were used for older pigs. Please clarify what collection was made for each phase.

Line 215: Does this indicate that only fecal samples were analyzed for Lactobacillaceae?

Line 269: was crossfostering performed before sample collection for the initial exposure phase?

For 25 pigs, how many of them were crossfostered?

Line 277: What were the statistics analysis results?

Line 307: How about good2?

Line 310: Poor1 didn’t express L. vaginalis.

Line 334: Were results from those cross fostered differ from those were not? If there are, it indicated the evident of confounding effect. Was cross fostering also performed in the other farms?

Line 373: Please provide the data in supplemental.

Line 375: Since the different was only showed in initial exposure phase, it is no need to present Fig8a.

Line 385: Please correct P=0.000. Also space should be available before and after equal sign, P = 0.000, and P should be italic.

Please being consistent with “development group” throughout the whole document.

Line 404: Was any of these five pigs Cross fostered? Were other piglets also express these two bacteria?

Line 428: what is the explanation for the contrary observation?

Line 457: Was there any difference between farms?

6. PLOS authors have the option to publish the peer review history of their article (what does this mean?). If published, this will include your full peer review and any attached files.

Reviewer #1: No

Reviewer #2: **Yes: **Tsungcheng Tsai

---

## [Author Response · Author response to Decision Letter 0]

25 Mar 2024

Point-by-point response to reviewer comments

Thank you for the time and effort dedicated to providing feedback on our manuscript. We are grateful for the insightful comments and valuable improvements to the manuscript. We have incorporated the suggestions made by the reviewers. The changes are made visible with track changes within the manuscript. Our corrections refer to the new line numbers of the revised manuscript with track changes.

Editor comments:

Please provide much more details about the experiment to allow the reviewers to determine if the data is appropriately sound for publication in PloS one. Make sure to address all comments exhaustively.

Response – Thank you. We have carefully considered each comment and revised the manuscript accordingly.

Reviewer comments:

Reviewer #1: The manuscript attempts to describe an evaluation of microbial diversity in growing pigs with different growth performance. However, there are many shortcomings with the manuscript that need to be addressed and considered before further evaluation of the manuscript.

Lines 50-102: The Introduction should be shortened, targeted, and concluded with focused objective statements.

Response – Thank you for the comment. The introduction has been revised and shortened.

Lines 61-65: should be moved to Discussion

Response – We have considered this carefully, however, as our sampling times are based on the phases as described here, we feel that lines 61-65 is the appropriate place to introduce the microbial development phases. We think that the reader can get a better picture about the phases and suggest that we could leave this part in the introduction. 

Lines 76-84: should be moved to Discussion

Response – This paragraph has been removed and the remaining sentence in lines 80-83 has been combined to the previous paragraph. The remaining sentence serves as an introduction to Lactobacillaceae. 

Line 118: How many piglets per site? What were the sow production numbers (e.g. total born, still born)?

Response – The missing information has been added as Table S1 in the supporting information and referred to in lines 116-119.

Line 127: How far were piglets transported?

Response – Weaning units were situated within the piglet producing farms. This has been clarified in lines 136-137 as follows “After weaning at approximately four weeks of age, piglets were moved into the weaning unit on the same farm.”

Line 128: How many total pens in the weaning and finishing units?

Response – The corresponding two fattening farms for each piglet producing farm had space for 2200 and 2300 fatteners on Farm 1, on Farm 2 for 1100 and 3500 fatteners, and on Farm 3 for 1500 and 1700 fatteners. On finishing farms, the arriving pigs were grouped together with approximately 20 pigs per pen but as the pigs grew some pigs were moved to other pens to give the growing pigs more space. Towards the end of finishing there were less than 20 pigs in one pen. In average, throughout the finishing period there were 15 pigs in a pen, resulting the finishing farms having 147 and 154 pens for Farm 1, 74 and 234 pens for Farm 2, and 100 and 114 pens for Farm 3. With some additional empty pens were available for managing sick animals and possible overcrowding. 

Piglet producing farms 1-3 had approximately 96, 192, and 144 pens in the weaning units, respectively. Including some empty pens for sick animals etc.

Line 129: Did one weaning unit house all the pigs?

Response – On each farm the weaning unit housed all the weaned pigs. The weaning units were a part of the piglet producing farm building. 

Line 129: Were the finishing farm groups similar to the weaning unit groups or completely random?

Response – Finishing farm groups were random, since caretakers mixed pigs into groups upon arrival.

Line 137: Was feed tested for microbial contribution to the results?

Response – The farms are a part of slaughterhouse contract breeders with general feeding instructions. The feed formulation did not contain any added microbes and was therefore not tested in this study. A general diet composition has been added to the text in lines 147-150 as follows “On all farms homegrown grains (barley, wheat, oats) were mixed with a commercial protein source (barley-based protein feed or soy) and concentrate feed. All except suckling piglets were liquid fed and a mix of organic acids with a formic acid base was added to the liquid feed mixture for preservation.”

Line 138: What was the method and purpose of heat treatment?

Response – First feed for suckling piglets is a commercial feed that is heat treated (heated to a minimum of 81 degrees) at the feed factory to eliminate possible contamination of feed and ensure the safety of piglets. A clarification has been added in line 151-152.

Line 147: Is it correct that only ~5 piglets were born to each sow? Further on, only 25 piglets are considered, so the discrepancy must be explained.

Response – Thank you for the comment, this part was indeed written very confusingly, we apologize for that. On average 13.4 (Farm 1), 18.8 (Farm 2), and 15.0 (Farm 3) piglets were born to the sows tracked during one week on the farms. Thus, altogether 2366 piglets were born and ear-tagged. Out of these pigs, 960 (the three largest/smallest from each litter) were included in this study for follow-up and fecal sampling from birth to slaughter. After slaughter and evaluation of all collected data, criteria for forming the three developmental groups were created and 25 pigs met these criteria. The fecal samples of these 25 pigs and their dams were chosen for analysis. This was clarified in the text in lines 160-164, in table S1 in the supporting information, and in lines 190-194.

A further limiting factor for the number of samples was the sequencing process, which for reliable comparison of samples requires all samples to be analyzed at the same time. The lab could process only 96 samples per sequencing run. 

Line 148: How were pigs selected and identified?

Response – All dams giving birth a week before Check-up 1 were tracked and all piglets born during this week were ear-tagged at birth for individual identification (364, 1461, and 541 pigs respectively on Farms 1-3). Lines 160-164 have been revised for clarification.

Line 150: What time points were evaluated?

Response – For each of the three phases (initial exposure, transitional, and stable phase) one timepoint and corresponding fecal sample was evaluated. The justification for choosing these phases is presented in the introduction and in Figure 1. For the initial exposure phase, the time point was during the tracked birth week on the farms (Check-up 1, pigs were 2-6 days of age). For the transitional phase the timepoint was around weaning (Check-up 2, pigs were 23-42 days old), and for the stable phase it was at the end of the fattening period (Check-up 5, pigs were 133-149 days old). Timepoints and fecal sampling points are shown in Figure 2. A clarification was added to the text in lines 166–167. 

Line 151: Birth weight and current weight?

Response – Pigs were weighed only during birth week. Wording “during birth week” was added to the text in lines 168-169. Carcass weight was received from the slaughterhouse post slaughtering. 

Line 152: Were pigs selected for and marketed based on days on feed or body weight?

Response – Pigs were marketed based on body weight (around 120 kg live weight at slaughter). According to the selection criteria of the study, pigs were eligible for the study, if they grew either + or - 1/2x the farm average ADG. 

Line 154: Provide some (not all) details in the text for the reader.

Response – The text was revised, and details were added in lines 167-176.

Line 161: insert "during lactation" after "information"

Response – Information was collected throughout the lives of the pigs and therefore “during lactation” does not fit in line 190.

Line 162: delete "These criteria are summarized in"

Response – “These criteria are summarized in” was deleted.

Line 163: Why were only these pigs selected?

Response – For a manageable number of samples, three largest and three smallest pigs from each litter were selected to follow-up on their growth until slaughter (n=960). After slaughter and evaluation of all collected data, criteria for forming the three developmental groups were created and 25 pigs met these criteria. Inclusion criteria was availability of fecal samples (piglets and their dams) and antimicrobial medication information, and growth parameters (Table 1) leaving with the 25 piglets and their dams. 

Line 163: Does that mean at least one pig from each litter?

Response – Yes, all pigs, except for 2, were from different litters. 

Line 165: When did these pigs die?

Response – They died at the ages of 2, 28, 59, and 69 days. A clarification was added in the text in lines 194-195.

Line 165: Did you confirm the reason was injury?

Response – Unfortunately all farms did not collect information on all death reasons of animals even though they were expected to. Therefore, reasons of death for three of the PrematureDeath pigs are unknown, and the fourth one died due to diarrhea.

Line 175: How are the sampling dates considered in the statistical analysis?

Response – The data was aggregated into the three time points that correspond to approximate sampling time as described in lines 166-181 in Materials and Methods (see also Figure 2). The dates were not considered in the analysis.

Line 253: There is no indication how the potential influence of different farms was considered in the analysis. How was each farm tested as a separate block? How was parity influence considered? What are the performance metrics considered and how were they evaluated?

Response – Farm was not considered in the analysis. The aim was to isolate pig-derived Lactobacillaceae from the same slaughterhouse contract farms. The farms were chosen according to similar pig management and farm size (piglet production farms with 1000-1300 sows). Fecal samples were analyzed from 25 pigs of which 3 were from farm 1, 5 from farm 2, and 17 from farm 3. This does not invite for statistical analysis between farms. 

Parity influence was not taken into consideration, because five out of 24 sows were of first parity. Their offspring were in all three development groups: one in group good, 3 in group poorly, and 1 in group PrematureDeath. 

The 25 pigs were selected retrospectively based on ADG, antimicrobial treatments, and meat inspection results out of 960 pigs. For this reason, we could not include performance metrics in this study. Regarding the next study with a larger study group, performance metrics should be taken into consideration.

Line 306: Was this pig considered in further analysis?

Response – Pig Good8 was considered in further analysis even though no culturable Lactobacillaceae were detected in the transitional phase sample. Lactobacillaceae were detected in the initial exposure phase sample as well as in the stable phase sample. 

Line 401: What are other probable sources?

Response – Other probable sources for Lactobacillaceae would be the environment. 

Line 418: Were any surfaces swabbed in this study? How were the rooms cleaned and disinfected before farrowing?

Response – No surfaces were swabbed. The rooms were cleaned with a pressure washer between batches. 

Line 418: Why were cross-fostered pigs chosen and not excluded? 

Response – We had no possibility to influence the management on the farms. All farms were commercial farms with their own practices and management routines. Given the choice, we would have excluded cross-fostered animals. However, cross-fostering was very common on these farms (around 70% of piglets were cross-fostered, estimation of the researchers). Since the study was conducted on commercial farms, excluding cross-fostered piglets would not describe the real microbial situation in the guts of commercial pigs.

Line 444: Why were medicated pigs chosen and not excluded?

Response – The majority of medicated pigs reached slaughter age. In our other manuscript, which is also under revision, we saw decreased growth in medicated pigs indicating these pigs filling the criteria of poor growth for this study. 

Lines 486-487: This is a major shortcoming of the manuscript. Diet composition is known to influence the microbiome. Lack of information such draws serious questions about the validity of the study results.

Response – Thank you for the relevant comment. This study was conducted to follow up on the growth rate of the pigs in reflection to their gut Lactobacillaceae. The source of the Lactobacillaceae was secondary but discussed based on the literature. It is known, that early-life microbiome, including Lactobacillaceae, received from the dam shapes the core microbiome of pigs. Feed (other than dam milk) is more of a secondary influence at the beginning. When undisturbed (e.g., no administration of antimicrobials) the diversity of the core microbiome remains throughout the animals life. If the diversity however is disturbed, it will not recover. This core microbiota is the basis of health and of interest to us in this study. 

A secondary microbiota develops after weaning and this microbiota is very adaptable to influences and also very recovering. Diet and the environment influence the secondary microbiota in short-term meaning; as long as microbes are provided (through feed or the environment) they can be found in the gut, but as soon as the provision ends also microbes disappear from the gut. In this study, pigs received the same diet throughout each farm and therefore the effect of the microbiome is seen in all animals. 

For this study we aimed to identify the core microbiome in correlation with pig growth. The text in the limitations paragraph has been revised accordingly in lines 545-550.

Figures: All figures should include all the information needed to stand alone.

Response – Legends of figures have been revised to meet the standards of the journal. 

Reviewer #2: Author studied Fecal microbiota profiles of growing pigs and their relation to growth performance.

Line 113: since there were only three farms, please note the number of sows and finisher pigs from each farm.

Response – Thank you for the comment. Numbers of sows and finisher pigs for each farm have been added in lines 116-123.

Line 118 please provide the size of pen.

Response – Pen sizes were according to Finnish standards. Farrowing pens: 4,5 m2. Space allowance for pigs in the weaning unit and on the finishing farm increase from 0.4 m2 for 30 kg pigs to 0.9 m2 for 120 kg pigs. This clarification has been added to the text in lines 125 and 139-142. 

Line 122 “The pens were never completely clean.”

Were pens cleaned and disinfected before introducing the next group of sows?

Were sows from all 3 farms received the straw before parturition?

Response – Yes, pens were cleaned with a pressure washer between batches. This was clarified in the text in lines 130-132.

On all three farms sows received some straw before parturition as nest-building material.

Line 123 What was the rule used for crossfoster?

Response – Since all three farms were commercial farms, the caretakers were responsible for animal management, including cross-fostering. All farms had their own practices and management routines. Piglets were regrouped to sows based on the judgement of the caretakers within a few days of farrowing. A clarification was added to the text in line 132.

Line 132 to 135: Were these practices true for all 3 farms?

Response – On all three farms pigs received chains and/or pieces of wood for enrichment. Also, sawdust was introduced in the weaning unit and sawdust/straw/peat were added by the caretakers on a regular basis in weaning units and finishing farms regardless of the farm. Lines 142-143 were revised accordingly.

Line 138: Is there scientific evidence that can confirm no probiotics are being added in any stage of production from all three farms? Since diets have greatly impacted the microbiomes of pigs, please provide diet compositions.

Response – The farms are a part of slaughterhouse contract breeders with general feeding instructions. The feed formulation did not contain a

---

## [Decision Letter · Decision Letter 1]

11 Apr 2024

Fecal microbiota profiles of growing pigs and their relation to growth performance

PONE-D-24-03235R1

Dear Dr. König,

We’re pleased to inform you that your manuscript has been judged scientifically suitable for publication and will be formally accepted for publication once it meets all outstanding technical requirements.

Kind regards,

Franck Carbonero, PhD

Academic Editor

PLOS ONE

Additional Editor Comments (optional):

Please consider Reviewer 2's questions and possibly make minor changes addressing them during the proofing stage.

Reviewers' comments:

Reviewer's Responses to Questions

**Comments to the Author**

1. If the authors have adequately addressed your comments raised in a previous round of review and you feel that this manuscript is now acceptable for publication, you may indicate that here to bypass the “Comments to the Author” section, enter your conflict of interest statement in the “Confidential to Editor” section, and submit your "Accept" recommendation.

Reviewer #1: All comments have been addressed

Reviewer #2: (No Response)

2. Is the manuscript technically sound, and do the data support the conclusions?

Reviewer #1: (No Response)

Reviewer #2: Yes

3. Has the statistical analysis been performed appropriately and rigorously? 

Reviewer #1: (No Response)

Reviewer #2: Yes

4. Have the authors made all data underlying the findings in their manuscript fully available?

Reviewer #1: (No Response)

Reviewer #2: Yes

5. Is the manuscript presented in an intelligible fashion and written in standard English?

Reviewer #1: (No Response)

Reviewer #2: Yes

6. Review Comments to the Author

Reviewer #1: (No Response)

Reviewer #2: Line 147: What were the pig's selection criteria? How many piglets per dam were selected? What is the dam distribution for each farm?

Response – Parturitions were tracked on all three farms for one week. During this time on Farm 1 30 sows gave birth to 364 piglets, on Farm 2 91 sows gave birth to 1461 piglets, and on Farm 3 39 sows gave birth to 541 piglets. All piglets born during this week were ear-tagged at birth for individual identification. For this study, from each litter the three largest and three smallest pigs were regarded (n=960). From these pigs fecal samples and information was collected from birth until slaughter.

The text has been revised in lines 160-164 to express this more clearly. Also, Table S1 was added to the supporting material to provide additional information on farms and sows.

Q: It is surprising that the mortality of pigs from this trial, especially those with the most miniature pigs. what is the typical mortality rate in these systems? Were medication diets and zinc and copper offered in these systems?

ine 163: How was the gender distribution between the three categories and the number of pigs distributed from each farm?

The number of observations for the Premature death group is very low. At which stage of production were these 4 pigs dead?

Response – Thank you for bringing this up. Gender was not an inclusion criterion in this study. We did not consider gender as a criterion because throughout the study pigs have not reached puberty and therefore gender specific hormones have likely not yet influenced the microbiota. Three pigs out of 25 were female. Gender distribution has been added to Table 2, which also shows distribution of pigs from each farm.

Out of the pigs in the PrematureDeath group one died during suckling, one around weaning and the remaining two before transport to finishing farm at 10 weeks of age. This has been added to the text in lines 194-195.

Q: What BW average were pigs marketed? Pigs typically reach their puberty around 23 to 26 weeks of age.

Line 457: Was there any difference between farms?

Response – The farms are a part of slaughterhouse contract breeders with general animal husbandry guidance. Apart from this, one farm was located in south-western Finland, whereas the other two were a little further in the north. One farm had a lower medication threshold and another farm had better equipped creep areas for the nursing piglets. Otherwise, farms were quite similar in their management, routines, and environments (farm design etc.).

Q: Was there a farm effect on the abundance of Lactobacillaceae?

7. PLOS authors have the option to publish the peer review history of their article (what does this mean?). If published, this will include your full peer review and any attached files.

Reviewer #1: **Yes: **Ryan Samuel

Reviewer #2: **Yes: **Tsungcheng Tsai

---

## [Editor Report · Acceptance letter]

26 Apr 2024

PONE-D-24-03235R1 

PLOS ONE

Dear Dr. König, 

I'm pleased to inform you that your manuscript has been deemed suitable for publication in PLOS ONE. Congratulations! Your manuscript is now being handed over to our production team.

Kind regards, 

on behalf of

Dr. Franck Carbonero 

Academic Editor

PLOS ONE